# The Minimally Invasive SERI Osteotomy for Pediatric Hallux Valgus

**DOI:** 10.3390/children10010094

**Published:** 2023-01-02

**Authors:** Gino Rocca, Angela De Venuto, Antonio Mazzotti, Simone Ottavio Zielli, Elena Artioli, Lorenzo Brognara, Francesco Traina, Cesare Faldini

**Affiliations:** 1Pediatrics Orthopedics and Traumatology, IRCCS Istituto Ortopedico Rizzoli, 40136 Bologna, Italy; 2Azienda Ospedaliera Maggiore della Carita di Novara, 28100 Novara, Italy; 3IRCCS Istituto Ortopedico Rizzoli, 1st Orthopaedics and Traumatology Clinic—University of Bologna, 40136 Bologna, Italy; 4Department of Biomedical and Neuromotor Sciences (DIBINEM), Alma Mater Studiorum University of Bologna, 40123 Bologna, Italy; 5Ortopedia-Traumatologia e Chirurgia Protesica e dei Reimpianti d’anca e di Ginocchio, IRCCS Istituto Ortopedico Rizzoli, 40125 Bologna, Italy

**Keywords:** adolescent, juvenile, hallux valgus, SERI, distal linear metatarsal osteotomy

## Abstract

Hallux valgus (HV), one of the most common forefoot pediatric pathologies, is characterized by lateral deviation of the big toe and medial displacement of the first metatarsal bone. Different surgical techniques have been described to correct the deformity, but no consensus has been reached regarding the best surgical option. The aim of this retrospective study was to report the results of the SERI technique in 58 pediatric HV of 45 consecutive patients. Pre- and postoperative VAS, AOFAS score, HVA, IMA and DMAA were collected. Preoperatively 3 patients (5.2%) had a mild deformity, 52 patients (89.6%) had moderate deformity and 3 patients (5.2%) had severe deformity according to Coughlin et al. Mean VAS score decreased from 5.2 ± 2.2 preoperatively to 0.8 ± 0.4 postoperatively. Mean AOFAS score improved from 68.1 ± 6.8 (range 59–75) preoperatively to 96.3 ± 3.2 (range 88–100) postoperatively, mean HVA reduced from 28.4° preoperatively to 13.2° postoperatively, mean IMA decreased from 15.2° preoperatively to 9.5° postoperatively (*p* < 0.01); mean DMAA decreased from 13.7° preoperatively to 8.2° postoperatively (*p* < 0.01). SERI technique showed satisfactory results when treating mild to severe pediatric HV deformity. No major complications were reported.

## 1. Introduction

Hallux valgus (HV) is one of the most common forefoot pediatric pathologies, with a prevalence of about 8% [1,2]. HV is characterized by lateral deviation of the big toe and medial displacement of the first metatarsal bone.

Etiology is undefined, although female sex, maternal familiarity, concomitant flat-foot deformity, relatively long or varus first metatarsal bone, ligament laxity, congruence and hypermobility of the metatarsophalangeal (MTP) joint are known risk factors [3,4].

It has been hypothesized that an oblique conformation of the wedge-metatarsal joint favors a hypermobility of the first metatarsal base in the medial sense, with a consequent increase in the inter-metatarsal angle (IMA). Hindfoot over-pronation also induces a longitudinal rotation of the first ray, with consequent malalignment of the flexor kinematic system of the first ray, and progressive worsening of the MTP joint deformity [3,4,5].

Pediatric HV treatment can be conservative or surgical. Surgical treatment is usually indicated in the case of symptomatic HV, when bracing or orthotic management are not effective [4,6].

Many surgical procedures have been described in the literature, although no consensus has been reached regarding the best surgical option [4,7,8,9,10,11]. This issue is particularly relevant in pediatric patients, in which the best time for surgery and the most appropriate techniques are still a matter of discussion [3,12].

The SERI technique, acronym for Simple, Effective, Rapid and Inexpensive, is a well established surgical technique for mild to moderate HV, with excellent clinical radiographic results reported [11,13]. Nevertheless, only a few studies reported its application in pediatric patients [14,15].

The aim of this retrospective study was to report the results of the SERI technique in a consecutive series of pediatric patients with HV.

## 2. Materials and Methods

In accordance with the Declaration of Helsinki, informed consent of the patients and their parents was collected before the surgery. The patient and his family were extensively informed about the surgical procedure, and the associated risks and benefits. All patients and parents signed a dedicated informed consensus.

A query of the institutional database was performed, to identify all patients diagnosed with HV, and treated with SERI.

Inclusion criteria were: (1) juvenile and adolescent patients between 8 and 17 years old at the time of the surgery, (2) mild to severe HV (HVA > 16° and IMA > 10°), (3) joint congruence associated with the absence of joint degeneration sign, (4) medial bursitis of the first MTP joint with shoe wearing limitation, (5) minimum follow-up of 12 months and (6) physiological metatarsal formula [16,17]. We only considered patients with a “zero-plus” first metatarsal, meaning a first metatarsal bone as long as or slightly longer than the second metatarsal bone.

Established exclusion criteria were: (1) rheumatoid arthritis or other juvenile chronic inflammatory joint diseases, (2) neurological disorders and (3) any previous foot surgery.

Patients under the age of 14 underwent general anesthesia, while patients over the age of 14 were managed with deep sedation and spinal anesthesia.

Perioperative prophylaxis with Cefazoline, or Clindamycin in allergic patients, was carried out before positioning an inflatable tourniquet.

Before surgical incision, a varus stress to the first metatarsophalangeal joint was applied for 20–30 s in order to lengthen the lateral capsular structures and the adductor hallucis longus. Correction of hallux valgus with the SERI minimally invasive technique, acronym for Simple, Effective, Rapid, Inexpensive, involves a distal linear osteotomy practiced approximately 1 cm proximal to the prominence of the first metatarsal head, as previously described [11]. A mini-incision is performed at the level of the neck of the first metatarsal, in a central position in reference to the dorso-plantar plane. A too distal osteotomy, below the vascular peduncle, predisposes to metatarsal head necrosis and sesamoids arthritis, while a too proximal osteotomy increases the risk of delayed union.

After gently detaching the periosteum, the bone surface is exposed. Osteotomy inclination should be perpendicular to the long axis of the second metatarsal in a coronal plane, to preserve the first metatarsal length.

For temporary fixation, a 2 mm K wire was inserted, through the incision, inside the soft tissues, parallel to the bone, proximally-distally along the longitudinal axis of the big toe. The K wire should emerge about 5 mm below the nail edge, at the apex of the big toe, and is then retracted until it reaches the osteotomy line. Using a special grooved probe as leverage, the K wire is placed into the diaphyseal canal, and the metatarsal head is displaced. Different modalities of K wire location during the insertion allows dorsal or plantar displacement of the metatarsal head [13].

Medial prominence bone spur resection was not consistently performed in case of skeletally immature patients without clear impingement with soft tissues.

After surgery, a static adherent forefoot dressing was applied for 15 days.

As regard to postoperative management, deambulation was allowed the first postoperative day using a Baruk shoe. Forefoot dressing was renewed after 15 days. The K wire was removed at the 40th postoperative day, and a last corrective bandage was applied and kept in place for 7–10 additional days, during which a gradual abandonment of the Baruk shoe was recommended in favor of comfortable shoes.

The usage of a full time postoperative toe alignment splint was encouraged for the following 40 days, and then worn only at night for two additional months.

Clinical evaluation of all patients was performed preoperatively, 6 months after the surgery and at the last follow-up by two orthopedic surgeons.

Clinical evaluation was performed through a physical examination and patient related outcome scores (PROMs).

The routine preoperative physical evaluation included the passive reducibility of the deformity, the soft tissue evaluation and any concomitant foot deformity.

The PROMs deployed were the Visual Analogue Scale (VAS) and the American Orthopedic Foot and Ankle Society Score (AOFAS). The patient was assigned a score from 0 to 100 based on the analysis of subjective and objective parameters such as pain, function, first ray clinical axis, footwear limitation, sport activity, stability of the metatarsophalangeal and interphalangeal joint, plantar callosity development [14].

Radiographic evaluation with weight-bearing dorsoplantar and lateral view radiographs was carried out preoperatively and at the 6 months follow up. Through radiographic examination preoperative and postoperative HVA, IMA and DMAA were evaluated.

The HVA (Hallux Valgus Angle) is represented by the angle between the long axis of the first metatarsal and the first proximal phalanx, and is normal if less than 15° (10°–15°); the IMA (InterMetatarsal Angle) is the angle between the long axis of the I and II metatarsal, it is usually less than 9° (7°–9°); the DMAA (Distal Metatarsal Articular Angle) or PASA (Proximal Articular Set Angle), normally 8°–10°, is the angle between the axis of the first metatarsal and a line perpendicular to the distal articular surface of the head of the first metatarsal.

Three degrees of HV severity were identified according to Coughlin [18,19,20]:Mild: HVA between 15° and 19°, IMA between 9° and 13°, sesamoid subluxation in anteroposterior X-ray less than 50%Moderate: HVA between 20° and 40°, IMA between 14° and 19° and sesamoid subluxation in anteroposterior X-ray between 50 and 75%Severe: HVA greater than 40°, IMA greater than 20° and sesamoid subluxation in anteroposterior X-ray greater than 75%.

An independent researcher performed the statistical analysis using the Jamovi Software (the Jamovi project—Jamovi 2.2.5).

Mean and standard deviation of continuous variables and clinical scores were calculated. The analysis of variance for repeated measures was used to evaluate the trend of the clinical scores during the follow-up period.

Single-way analysis of variance was used to evaluate the difference between inter-group values when the variables followed a normal distribution, applying the Student’s *t*-test. Otherwise, the non-parametric Wilcoxon rank test was used.

The trend of the clinical scores over the course of the follow-up period was assessed using the analysis of variance for repeated measures. When the variables had a normal distribution, the Student’s *t*-test was performed to examine the difference between inter-group values using the single-way analysis of variance. In all other cases, the Wilcoxon rank test was applied. To determine the statistical significance of the preoperative, postoperative clinical scores, and radiographic data, statistical tests were run. *p* 0.05 was taken into account as significant for all tests.

Intra and postoperative complications were recorded for each patient. At the final follow-up visit available, an HVA > 16° was considered an under correction, while an HVA < 0° was considered a case of iatrogenic hallux varus.

Overall complications were registered during follow-up.

## 3. Results

A total of 57 consecutive patients with mild to severe juvenile or adolescent hallux valgus were treated from December 2016 to December 2020.

After screening for inclusion and exclusion criteria, 46 patients were included in the study, for a total of 59 feet. One patient was lost at follow up. A total of 58 feet in 45 consecutive patients were retrospectively reviewed. The population was composed of 40 females and 5 males, with a mean age at the time of surgery of 12.5 ± 2.62 years (range 7–17) (Figure 1).

Last average follow-up visit was at 20.1 months (12–41 months).

### 3.1. Clinical Results

Mean VAS score decreased from 5.2 ± 2.2 preoperatively to 0.8 ± 0.4 postoperatively.

Mean AOFAS score significantly improved from 68.1 ± 6.8 (range 59–75) preoperatively to a postoperative mean value of 96.3 ± 3.2 (range 88–100) at last follow-up (*p* < 0.01) (Table 1) (Figure 2).

### 3.2. Radiological Results

Preoperatively, 3 patients (5.2%) had a mild deformity, 52 patients (89.6%) had moderate deformity and 3 patients (5.2%) had severe deformity according to Coughlin et al.

Mean HVA reduced from 28.4° preoperatively to 13.2° postoperatively (*p* < 0.01); mean IMA decreased from 15.2 preoperatively to 9.5 postoperatively (*p* < 0.01); mean DMAA score decreased from 13.7 preoperatively to 8.2 postoperatively (*p* < 0.01) (Table 2).

Fifty-five patients (94.8%) had no deformity at the final follow-up, while three patients presented under correction of the HV, with mild deformity (16° < HVA < 19° and 9° < IMA < 13°) (Table 3) (Figure 3).

### 3.3. Overall Complications

No delayed wound healing, clinical signs of infection or inflammatory reaction to the K-wire at the exit point were observed. No cases of sensory deficits, nonunion, deep infections, avascular necrosis of the metatarsal head or transfer metatarsalgia have been recorded.

## 4. Discussion

The purpose of the study was to report the results of the SERI technique in pediatric patients with mild to severe HV.

As described in the literature, a treatment with conservative measures, until after skeletal maturity, is recommended [21].

In the case of symptomatic HV, bracing or orthotics are available options, although they did not show the prevention of deformity progression. Moreover, it is uncommon for the juvenile and adolescent population to consistently wear orthoses or bracing [4].

When pediatric HV remains symptomatic despite conservative therapy, surgical treatment should be considered [4].

Many surgical techniques have been described, with a trend to prefer distal minimally invasive techniques [7,8,14,22,23,24,25]. When approaching a pediatric patient with open physes, distal metatarsal osteotomies are preferred, since first metatarsal osteotomies would risk violating the area of the growth plate [4].

Numerous different variations of distal metatarsal osteotomies have reported favorable outcomes. However these techniques have been described for mild to moderate deformities only, and do not allow rotational correction [24,25].

SERI technique has proven to be a safe, effective and reproducible technique to treat mild to moderate HV in the adult population in the last 25 years [11,13].

Numerous different variations of distal metatarsal osteotomies have reported favorable outcomes. However these techniques have been described for mild to moderate deformities only, and do not allow rotational correction [22,23].

SERI technique has proven to be a safe, effective and reproducible technique to treat mild to moderate HV in the adult population in the last 25 years [11,14].

The mini invasive approach and the distal linear osteotomy are the reasons why SERI can be considered a good option for juvenile and adolescent HV. Its effectiveness has already been proven in juvenile patients, even associated with subtalar arthroeresis for concomitant flatfoot or in a small retrospective case series [14,15]. Furthermore, the linear osteotomy allows for a rotational correction in case of overpronation of the first ray, in contrast to Chevron-type distal metatarsal osteotomies, with potential superior sesamoid reduction.

The surgical technique we applied is almost identical to the one described in adult patients [11]. However, in the pediatric patient, the bony prominence of the proximal stump of the osteotomy may not be removed, given the high intrinsic remodeling potential.

In the original description of the SERI, the forefoot dressing is maintained for 30 days, without being renewed [11]. In the pediatric population, patients tend to be more active and less compliant to postoperative recommendations, thus a dressing replacement at 15 days is recommended. The aim is to prevent dressing dislocation and to maintain a correct alignment of the first toe.

The deployment of the postoperative toe alignment splint after the removal of the dressing is another modification to the original postoperative protocol described in the SERI technique. This option represents an additional way to ensure the correction, enabling further remodeling of the osteotomy while the first ray is still kept well aligned.

Clinical and radiographical data showed good results in the majority of the patients.

Clinical satisfactory results were observed in 96.5% of the patients, comparable or even superior to the 70% of patients with excellent rating reported by Geissele et al. [26]. AOFAS score at the last follow up was 96.3, comparable to the 90.7 reported by Caravelli et al. [14], the 94.5 reported by Kraus et al. [7] or the 88.9 reported by Choi et al. [27].

No stiffness or pain in the first MTP joint was observed, although this condition is reported after surgery in an adult population [13]. A possible explanation could be the rare progression of the metatarsophalangeal osteoarthritis and the more elastic soft tissues condition in the pediatric population.

Radiographic correction was achieved in more than 94% of the treated patients. Average HVA reduction was 15.2°, average IMA decrease was 5.7° and average DMAA decreased was 5.5°. Only three patients (5.1%) reported under correction (HVA > 16°). All of them presented high preoperatory IMA values, which have been linked to an inferior postoperative correction [22], and despite the slight under correction, two out of three were satisfied by the results.

One of the main concerns regarding minimally invasive distal metatarsal osteotomy for HV is the loss of correction following the K wires removal [28]. As a matter of fact, Giannini et al. [29] and Maffulli et [30] al reported no statistical significant difference as regard clinical and radiological results comparing minimally invasive distal metatarsal osteotomy to traditional open surgical technique. Our results showed that at an average of 20.7 months after surgery, the correction obtained intraoperatively was maintained. These findings are comparable to those reported by other authors exploiting SERI technique for pediatric HV, describing the preservation of the correction at the last follow up [14,15,27].

A meticulous surgical technique, a proper postoperative forefoot dressing, together with the postoperative splint and the Baruk shoes employment, are key elements for the clinical and radiographic outcomes. This may explain the different postoperative results reported by a few authors [28].

Although the ideal treatment for pediatric HV is still a matter of debate, and conservative treatment should always be the first option, the numerous surgical techniques described to treat HV show the upgrowing interest towards this pathology.

The SERI technique, described in the 90s and already established in the adult population, still represents a good compromise thanks to the noncomplex surgical procedure (Simple), the satisfactory results (Effective), the short surgical timing (Rapid) and the hardware cost and availability (Inexpensive).

This study has some limitation. Mean follow-up was inferior to 24 months, so only premature relapse or complications could be identified. Furthermore, the retrospective nature of the study led to different follow-up periods during the last clinical and radiological assessment of the patients. Nonetheless, to our knowledge, this is the largest series of patient with juvenile HV treated with distal linear metatarsal osteotomy [7,14,26,31].

## 5. Conclusions

SERI technique showed satisfactory results when treating mild to severe pediatric HV deformity. No major complications were reported, and results were comparable to other well-established techniques reported in the literature. Further studies with prospective design and longer follow-up are required to confirm the results of this retrospective survey.

## Figures and Tables

**Figure 1 children-10-00094-f001:**
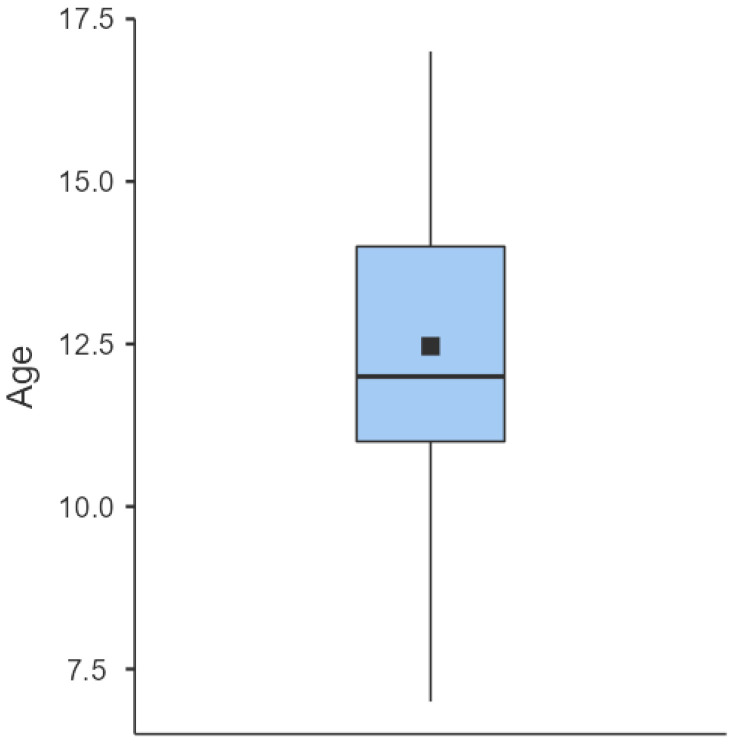
Age distribution box plot.

**Figure 2 children-10-00094-f002:**
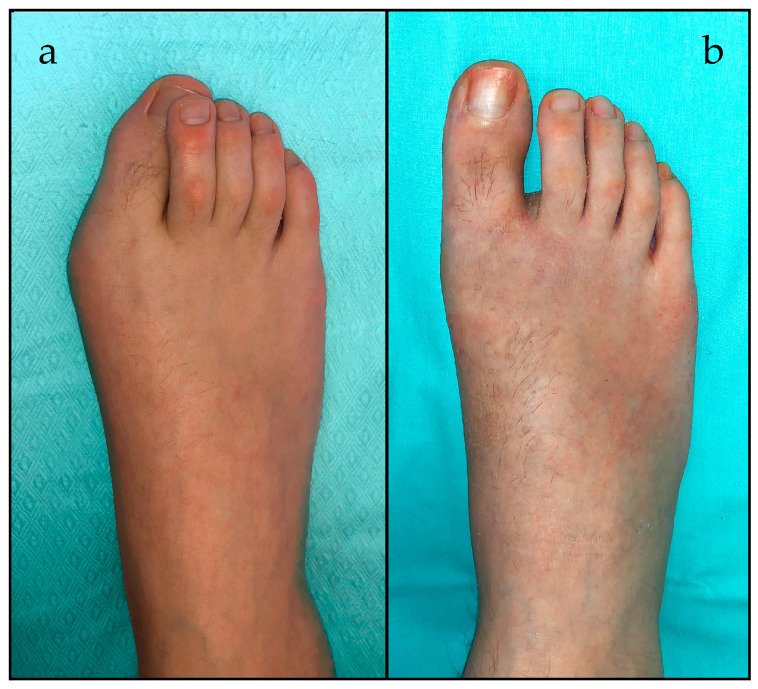
(**a**) Preoperative and (**b**) 12 months postoperative clinical documentation.

**Figure 3 children-10-00094-f003:**
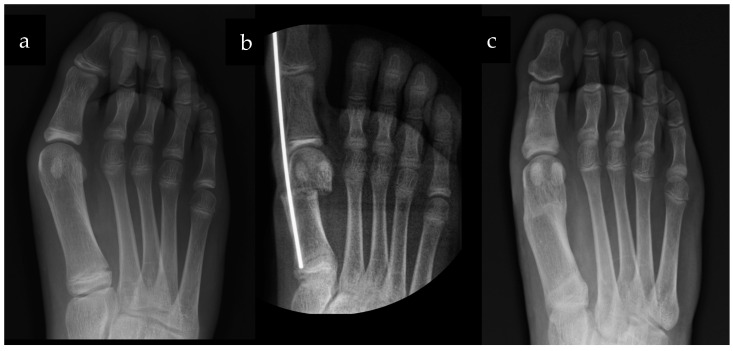
(**a**) Preoperative, (**b**) Intraoperative and (**c**) 12 months postoperative X-rays.

**Table 1 children-10-00094-t001:** Clinical results.

PROM	Preoperative Score Mean +/− SD (Range)	Postoperative Score Mean +/− SD (Range)
VAS	5.2 +/− 2.2 (4–8)	0.8 +/− 0.4 (0–3)
AOFAS	68.1 +/− 6.8 (59–75)	96.3 +/− 3.2 (88–100)

PROM = Patient Related Outcome Measure; VAS = Visual Analogue Scale; AOFAS = American Orthopaedic Foot and Ankle Score for Hallux Valgus; SD = Standard Deviation.

**Table 2 children-10-00094-t002:** Radiological results.

Radiological Parameters	Preoperative Angle Mean +/− SD (Range)	Postoperative Angle Mean +/− SD (Range)	*p* Value Wilcoxon Test	Degree Correction (°) Mean +/− SD
HVA	28.4 +/− 5.7 (17–42)	13.2 +/−2.4 (10–24)	<0.01	15.1 +/− 4.5
IMA	15.2 +/− 3.2 (10–22)	9.5 +/− 1.9 (6–20)	<0.01	5.6 +/− 3.1
DMAA	13.7 +/− 4.1 (5–22)	8.2 +/− 1.9 (5–12)	<0.01	5.5 +/− 3.1

HVA = Hallux Valgus Angle; IMA = InterMetatarsal Angle; DMAA = Distal Metatarsal Articular Angle.

**Table 3 children-10-00094-t003:** Pre- and postoperative deformities according to Coughlin classification.

Coughlin Classification	Preoperative		Postoperative	
	Number of Patients	Percentage	Number of Patients	Percentage
No deformity	0	0%	55	94.8%
Mild deformity	3	5.2%	3	5.2%
Moderate deformity	52	89.6%	0	0%
Severe deformity	3	5.2%	0	0%
Total	58	100%	58	100%

## Data Availability

The data presented in this study are available on request from the corresponding author. The data are not publicly available due to the Institute privacy policy.

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
