# Peer review of "The Minimally Invasive SERI Osteotomy for Pediatric Hallux Valgus"

_children, 2023, doi:10.3390/children10010094_

Round 1

Reviewer 1 Report

First of all, thank you for your work, surgical treatment is really rare in HAV in the pediatric population and with your study, the field of knowledge is expanded.

Just make some considerations that can improve the article:

-        Understanding that it is a retrospective study, I believe that the methodology is incomplete. The protocol established in data collection, pre and post surgery, must be clarified. The study is carried out for four years and we must ensure reliability in data collection, taking into account that we value subjective data, measurements are taken... I would like it to be clarified if this is collected by a single researcher... etc. provide sufficient data to take into account possible biases.

-        Were the parents or guardians informed of the use of the data and would the results be used for research purposes?

-        Can you clarify what is meant by the physiological metatarsal formula, can you explain it to me taking into account the length of the first and second metatarsals?

I share with you all the considerations regarding the limitations of being a retrospective study, we do not have long-term results and it would be very interesting to recruit patients to see results.

Author Response

First of all, thank you for your work, surgical treatment is really rare in HAV in the pediatric population and with your study, the field of knowledge is expanded.

Just make some considerations that can improve the article:

Q -   Understanding that it is a retrospective study, I believe that the methodology is incomplete. The protocol established in data collection, pre and post surgery, must be clarified. The study is carried out for four years and we must ensure reliability in data collection, taking into account that we value subjective data, measurements are taken... I would like it to be clarified if this is collected by a single researcher... etc. provide sufficient data to take into account possible biases. 

A - Thank you very much for your observation. The manuscript has been edited as follows:

Clinical evaluation of all patients was performed preoperatively, 6 months after the surgery and at the last follow-up by two orthopedic surgeons. [see line 107-108]

Q -        Were the parents or guardians informed of the use of the data and would the results be used for research purposes?

A - Thank you for your observation. The manuscript has been edited as follows:

The patient and his family were extensively informed about the surgical procedure, and the associated risks and benefits. All patients and parents signed a dedicated informed consensus. [see line 58-60]

Q -        Can you clarify what is meant by the physiological metatarsal formula, can you explain it to me taking into account the length of the first and second metatarsals?

A - Thank you for your observation. The manuscript has been edited as follows:

We only considered patients with a “zero-plus” first metatarsal, meaning a first metatarsal bone as long as or slightly longer than the second metatarsal bone. [see line 67-69]

I share with you all the considerations regarding the limitations of being a retrospective study, we do not have long-term results and it would be very interesting to recruit patients to see results.

A - Thank you for your appreciation. As highlighted in the manuscript it is in our interest to keep following these patients to gather long-term clinical and radiological information.

Reviewer 2 Report

First, I would like to express sincere gratitude to get the opportunity to review your manuscript.

The effort of the author is appreciated, as the topic is interesting and promising. Congratulation on your results. A great introduction to the subject of the manuscript. 

After assessing the manuscript, the following issues raised my concern or represent suggestions that could in my opinion improve the quality of the manuscript:

- Discussion section of the manuscript - please address the following: Implication and explanation of findings. Recommendation and future directions.

- “Antiseptic prophylaxis” to be changed maybe with a more appropriate term like perioperative prophylaxis. 

- If possible, could you provide some pre and postop X-rays?

Author Response

First, I would like to express sincere gratitude to get the opportunity to review your manuscript.

The effort of the author is appreciated, as the topic is interesting and promising. Congratulation on your results. A great introduction to the subject of the manuscript. 

After assessing the manuscript, the following issues raised my concern or represent suggestions that could in my opinion improve the quality of the manuscript:

Q - Discussion section of the manuscript - please address the following: Implication and explanation of findings. Recommendation and future directions.

A - Thank you very much for you appreciation and the valuable suggestions. The manuscript has been edited as follows:

A meticulous surgical technique, a proper post-operative forefoot dressing, together with the post-operative splint and the Baruk shoes employment, are key elements for the clinical and radiographic outcomes. This may explain the different post-operative results reported by a few authors [28].

Although the ideal treatment for pediatric HV is still a matter of debate, and conservative treatment should always be the first option, the numerous surgical techniques described to treat HV show the upgrowing interest towards this pathology.

The SERI technique, described in the 90s and already established in the adult population, still represents a good compromise thanks to the non-complex surgical procedure (Simple), the satisfactory results (Effective), the short surgical timing (Rapid) and the hardware cost and availability (inexpensive).  [See line 270-280]

Q - “Antiseptic prophylaxis” to be changed maybe with a more appropriate term like perioperative prophylaxis. 

A - Thank you for your suggestion, the manuscript has been edited as suggested:

Perioperative prophylaxis with Cefazoline, or Clindamycin in allergic patients, was carried out before positioning an inflatable tourniquet. [see line 75-76]

Q - If possible, could you provide some pre and postop X-rays?

A - Thanks for your suggestion, pre operative, amplioscopic and post-operative X-rays have been added to the document.

Reviewer 3 Report

Dear authors,

Thank you for the opportunity to review this very good paper.

In my opinion the article is ready to be publish.

Good luck!

Author Response

Thank you very much for your appreciation and your encouragement.